# OpenReview forum: "Token-Level Adversarial Prompt Detection Based on Perplexity Measures and Contextual Information"
_ICLR.cc/2025/Workshop/BuildingTrust — BuildingTrust_

### Official Review · Reviewer_VbD3 · 2025-02-22
**PPL-based detection method for adversarial prompt attacks**

**Rating:** 4
**Confidence:** 4

**Review:**

This paper proposes two methods for detecting adversarial prompt attacks in LLMs, one for binary classification (attacked/not attacked) and one more fine-grained. Both rely on the perplexity/probability of tokens to identify an attacked prompt.

Strengths: The paper is clearly written overall.

Weaknesses: The authors state that existing methods focus on detecting prompt attacks at the sequence level only, while their work is the first to focus on token-level detection. On the other hand, the perplexity of individual tokens is considered together with the perplexity of their surrounding tokens, since -- as the authors themselves state -- only looking at individual token perplexity might inaccurately flag rare tokens as attacks. It is unclear to this reviewer what the strategies proposed here achieve that is not also achieved by established methods, such as the perplexity sliding window proposed in [1]. Sliding windows of different token lengths (up to a sliding window of length=1) can capture different granularities, and this method measures both the severity of the PPL increase and also lends itself to binary classification (using a threshold).

[1] (Jain et al., 2023) Baseline defenses for adversarial attacks against aligned language models

---

### Official Review · Reviewer_gg8Z · 2025-02-24
**New perplexity-based design for detecting adversarial prompts, but narrowly focuses on GCG...**

**Rating:** 5
**Confidence:** 4

**Review:**

The paper proposes a token-level perplexity-based method to detect adversarial prompts. In particular, it focuses on the class of GCG-produced adversarial prompts (Zou et al., 2023) and integrates the contextual information in the design to avoid high false positive rates. The interesting part of this paper is the new probabilistic model for modeling the perplexity of each token, which formulates a new optimization problem with regularization terms of contextual coherence among adjacent tokens (Equation 1 and Equation 2).

The experimental results seem promising. However, given the existing broad literature on jailbreak attacks on LLMs, the paper narrowly focuses on GCG, a very early approach known to generate adversarial suffixes with high perplexity. I view this as a major positioning issue of this paper. The question is whether the proposed detection method is generalizable to achieve good performance for other types of jailbreak attacks. It is worth noting that there are automatic jailbreak attacks, like AutoDAN [1], PAIR [2], and many other manual methods that handcraft "natural" adversarial prompts, achieving much lower perplexity compared with GCG. The authors should apply their detection method to these stealthier jailbreak attacks for more comprehensive evaluations.

[1] Zhu et al., AutoDAN: Interpretable Gradient-Based Adversarial Attacks on Large Language Models, https://arxiv.org/pdf/2310.15140

[2] Dobriban et al., Jailbreaking Black Box Large Language Models in Twenty Queries, https://arxiv.org/pdf/2310.08419v3

---

### Official Review · Reviewer_nMvx · 2025-03-02
**A good paper with a few areas for improvement**

**Rating:** 7
**Confidence:** 3

**Review:**

# Summary
The paper introduces a novel approach for detecting adversarial prompts in Large Language Models (LLMs) at the token level. The authors propose methods that leverage both token perplexity and contextual information to identify specific tokens that are likely part of adversarial inputs.

Their technical contribution consists of two algorithms:
1. A discrete optimization approach
2. A probabilistic graphical model

# Strengths
* **Novel contribution:** The authors present a new and non-trivial adversarial prompt detection method, which they test on a range of models and performance metrics related to both sequence- and token-level adversarial prompt detection.
* **Clarity of writing and presentation:** The paper is well-structured and clearly written. The methodology is explained in a logical progression, and the mathematical formulations are presented with sufficient detail.
* **Practicality:** The authors highlight that their methods can be implemented with relatively limited computational demands and hardware requirements. They demonstrate that their algorithms can be implemented in linear time and note that even GPT-2-small, which can be run on a CPU, achieved perfect results at the sentence-level detection task.

# Weaknesses
* **Lack of validation set:** While the authors performed a grid search to determine optimal hyperparameter values for different models, they did not evaluate the performance of these hyperparameters on a separate validation set or investigate how well they transfer to significantly different datasets. This is an important consideration in assessing this method's usefulness for real-world applications.
* **Limited comparison to previous work:** The paper presents impressive results with their methods achieving perfect classification performance at the sequence level. However, it would be informative if they compared their results to existing sequence-level adversarial detection methods. Does their approach represent a significant improvement?
* **Insufficient motivation for token-level detection:** The authors acknowledge that perplexity-based methods for sequence-level detection of adversarial prompts are well-established. However, they provide limited motivation for why token-level detection is uniquely useful or important. The paper would be strengthened by including concrete use cases where identifying specific adversarial tokens provides practical advantages over sequence-level detection.
* **Typos:** (line 024) Additionaly -> Additionally,  (line 142) produce -> produces, (Table 3) Performance Performance -> Performance.

# Overall Assessment
This paper makes a valuable contribution to the field of LLM security by introducing token-level adversarial prompt detection methods. While the technical approach is sound and the results are promising, addressing the noted weaknesses would strengthen the work considerably. In particular, more rigorous validation and better motivation for token-level over sequence-level detection would help establish the usefulness of this approach.

---

### Decision · Program_Chairs · 2025-03-04

**Decision:**

Accept

**Comment:**

The paper presents a novel token-level adversarial prompt detection approach but lacks sufficient validation and comparison with existing methods. Its primary limitation is the narrow focus on GCG-based adversarial prompts without evaluating stealthier jailbreak techniques like AutoDAN or PAIR, which could challenge its generalizability.
Additionally, while the authors emphasize token-level detection, they do not convincingly demonstrate its advantages over sequence-level methods or established sliding window techniques. The absence of a validation set and limited discussion on hyperparameter transferability further weaken the study’s real-world applicability. Addressing these issues would significantly strengthen the paper's contributions.